# Bacteriophage-driven emergence and expansion of *Staphylococcus aureus* in rodent populations

Gonzalo Yebra[1], Daniel Mrochen[2], Stefan Fischer[3], Florian Pfaff[4], Rainer G. Ulrich[3], Kathleen Pritchett-Corning[5,6], Silva Holtfreter[2]*, J. Ross Fitzgerald[1]*

1 The Roslin Institute, University of Edinburgh, Edinburgh, United Kingdom, 2 Institute of Immunology, University Medicine Greifswald, Greifswald, Germany, 3 Institute of Novel and Emerging Infectious Diseases, Friedrich-Loeffler-Institut, Federal Research Institute for Animal Health, Greifswald-Insel Riems, Germany, 4 Institute of Diagnostic Virology, Friedrich-Loeffler-Institut, Federal Research Institute for Animal Health, Greifswald-Insel Riems, Germany, 5 Charles River, Research and Professional Services, Wilmington, Massachusetts, United States of America, 6 Office of Animal Resources, Harvard University Faculty of Arts and Sciences, Cambridge, Massachusetts, United States of America

* Silva.Holtfreter@med.uni-greifswald.de (SH); ross.fitzgerald@ed.ac.uk (JRF)

**Data Availability Statement:** Illumina sequence data is available in the European Nucleotide Archive under the BioProject PRJEB41130.

## Abstract

Human activities such as agriculturalization and domestication have led to the emergence of many new pathogens via host-switching events between humans, domesticated and wild animals. *Staphylococcus aureus* is a multi-host opportunistic pathogen with a global health-care and economic burden. Recently, it was discovered that laboratory and wild rodents can be colonised and infected with *S. aureus*, but the origins and zoonotic potential of rodent *S. aureus* is unknown. In order to trace their evolutionary history, we employed a dataset of 1249 *S. aureus* genome sequences including 393 of isolates from rodents and other small mammals (including newly determined sequences for 305 isolates from 7 countries). Among laboratory mouse populations, we identified multiple widespread rodent-specific *S. aureus* clones that likely originated in humans. Phylogeographic analysis of the most common murine lineage CC88 suggests that it emerged in the 1980s in laboratory mouse facilities most likely in North America, from where it spread to institutions around the world, via the distribution of mice for research. In contrast, wild rodents (mice, voles, squirrels) were colonized with a unique complement of *S. aureus* lineages that are widely disseminated across Europe. In order to investigate the molecular basis for *S. aureus* adaptation to rodent hosts, genome-wide association analysis was carried out revealing a unique complement of bacteriophages associated with a rodent host ecology. Of note, we identified novel prophages and pathogenicity islands in rodent-derived *S. aureus* that conferred the potential for coagulation of rodent plasma, a key phenotype of abscess formation and persistence. Our findings highlight the remarkable capacity of *S. aureus* to expand into new host populations, driven by the acquisition of genes promoting survival in new host-species.

**Funding:** The study was supported by grants to JRF from the Wellcome Trust (UK) (201531/Z/16/Z/WT) and the Biotechnology and Biological Sciences Research Council, UK (Institute Strategic programme; BBS/E/D/20002173/UKRI) and to SH by the Deutsche Forschungsgemeinschaft (GRK 2719/1). The funders had no role in study design, data collection and analysis, decision to publish, or preparation of the manuscript.

## Author summary

New pathogens typically emerge after host-switching events, threatening public health and food security. Here we used an evolutionary genomic approach to reveal the origins of *Staphylococcus aureus* in laboratory and wild rodent populations. *S. aureus* in laboratory rodents appears to have originated in humans and following host jump events has spread around the world via the distribution of mice for research purposes. Distinct clones of *S. aureus* widely distributed around Europe are associated with wild rodent populations. We found unique types of prophages in rodent *S. aureus* that have played a central role in their emergence and expansion including some with genes encoding putative virulence factors. In particular, we identified novel prophages and pathogenicity islands that conferred the potential for coagulation of rodent plasma, a key phenotype of abscess formation and persistence. Taken together, our findings highlight the remarkable capacity of *S. aureus* to expand into new host populations, driven by the acquisition of genes acquired via mobile genetic elements.

## Introduction

Domestication and human activities in general have had a profound impact on the emergence of new pathogens [1]. However, our understanding of the drivers for successful host-switching events and subsequent expansion and dissemination is very limited. *Staphylococcus aureus* is a major human and animal pathogen with a remarkable ability to colonise and infect a broad range of animal hosts including humans, livestock and wildlife [2,3]. The multi-host associations of *S. aureus* are related to its capacity to adapt to different ecological niches through an array of evolutionary genetic and functional mechanisms, including acquisition of mobile genetic elements (MGE) [4–8], allelic diversification of specific effectors [9,10], recombination [11] or even single point mutations [12]. Recent studies have revealed the existence of high-risk animal-associated clones of *S. aureus* that may represent an increased public health threat [13].

Laboratory mice are the most commonly employed species for *in vivo* modelling of *S. aureus* pathogenesis or for testing the efficacy of new vaccine candidates [14,15]. Typically, mice are challenged with human-adapted *S. aureus* isolates, such as USA300. However, it has been recently demonstrated that laboratory mice are natural *S. aureus* hosts, and that even specific-pathogen-free (SPF) mice are frequently colonized with *S. aureus* [16]. Natural colonization with *S. aureus* primes the adaptive immune system and thus likely impacts on the outcome of experimental infections or challenge experiments [16].

Multi locus sequence typing (MLST)-based genotyping of *S. aureus* isolated from laboratory mice in different countries has identified three main clonal complexes (CC) accounting for the majority of isolates: CC88, CC15 and CC1 [17]. While CC1 and CC15 are commonly associated with human infections, CC88 is rarely found in human populations in Western countries [18] and its origin remains unknown. Rodents in urban or human-influenced environments are typically colonised by *S. aureus* strains also found in humans and other animals such as CC8, CC398, and CC97, and frequently encode methicillin resistance [19–23]. Similarly, companion rodents such as guinea pigs, and companion rabbits are typically colonized by strains commonly associated with humans [24,25]. In contrast, wild rodents including mice, rats and squirrels are typically colonized with unique *S. aureus* lineages, including CC49, CC1956 and CC890 [26–28]. In particular, *S. aureus* CC49 (only sporadically found in humans) has been isolated from different species of wild mice in Germany [26], wild rats in Belgium [27] and red

squirrels in the UK [29,30]. The existence of CCs that are unique to wild rodents suggests host-specialization but the specific mechanisms that underpin this tropism are largely unknown.

Rodents are a major reservoir for bacterial zoonotic infections of humans [31]. It is imperative that we understand the origins of newly identified animal pathogens, to investigate the drivers of successful expansion into distinct host populations and to assess the public health threat. Here, we combine population genomic, phylodynamic and genome-wide association approaches along with functional analyses to trace the evolutionary history of rodent populations of *S. aureus* and to investigate the genetic and functional basis for host-adaptation. Our data reveal the time-frame for the emergence and international dissemination of the major rodent clones of *S. aureus*, and identify a novel family of bacteriophages that are associated with their ecological success.

## Methods

### Sequence dataset

Whole genome sequencing (WGS) of 340 previously described *S. aureus* isolates [17,26] was performed by MicrobesNG (Birmingham, UK) using the Nextera XT library prep protocol on a MiSeq platform (Illumina, San Diego, CA, USA). Sequences were deposited in SRA/ENA under the BioProject PRJEB41130. These isolates included i) 241 laboratory mice and 15 laboratory rat isolates from the US, Germany, Canada, France, New Zealand, the UK and Japan; ii) 43 wild mouse and vole isolates from Germany and the Czech Republic, iii) 5 shrew isolates from Germany; iv) 1 guinea pig isolate from Germany; and v) 35 human isolates belonging to CCs found among the rodent isolates (ST88, ST49, ST130 and ST8). No live participants were used in the current study. *S. aureus* isolates from laboratory mice were originally obtained as part of routine health surveillance, and occasionally from euthanized mice suffering from *S. aureus* infections (e.g. preputial gland adenitis). To avoid sample bias, we included a maximum of 4 isolates of the same *spa* type, barrier, mouse strain and time point in the study cohort.

To complement this dataset, *S. aureus* RefSeq genomes and short-read data deposited in SRA (**S1 Table**) were queried for sequences from rodents (mice, rats, squirrels, guinea pigs and others) (n = 64) and lagomorphs (rabbits, hares) (n = 25).

These genomes are listed in **S2 Table** with their available metadata. Short read sequences were adapter trimmed using Trimmomatic v0.36 [32] and *de novo* assembled using SPAdes v3.11.1 [33]. Genes were annotated using Prokka v1.13 [34]. All assemblies were typed using mlst v2.10 (https://github.com/tseemann/mlst) and sequence types (STs) were grouped into CCs using the eBURST algorithm in PhyloViz [35].

### Phylogenetic and phylodynamic analyses

In order to examine the phylogenetic relationships between the rodent isolates, we constructed a core genome phylogenetic tree using as background sequences a dataset of 798 *S. aureus* genomes representative of the global genetic diversity [5] using Parsnp v1.1.2 [36].

Monophyletic, rodent-specific clades were identified, and CCs with more than one rodent clade were selected for further phylodynamic analysis. Additional sequences belonging to the STs contained in each of these CCs of interest were obtained from SRA queried using Staphopia [37]. These were downloaded and assembled as explained above. CC-specific core genome alignments were created using snippy and snippy-core v4.1 (https://github.com/tseemann/snippy) using as reference for each dataset a closed genome belonging to that CC when available. Recombinant regions in those core genome alignments as detected by Gubbins v2.3.4 [38] were discarded. An initial tree of the recombinant-free single nucleotide polymorphism

(SNP) genome alignment was constructed using IQTree v1.6.12 [39] with the GTR+G4 model including an ascertainment bias correction with the number of invariable sites from the core genome alignment, and 1,000 rapid bootstraps. These trees were used to test the temporal signal of the dataset in TempEst v1.5.3 [40].

Bayesian phylogenetic analyses were performed using BEAST v1.9.0 [41], testing different models for nucleotide substitution (HKY and GTR), molecular clock (strict and uncorrelated lognormal relaxed) and demographic growth (constant, exponential and Bayesian skygrid). Each model combination was run for 100 million generations, with sampling every 10,000 and discarding the initial 10% as burn-in. Runs were compared via a marginal likelihood estimation (MLE) using path sampling and stepping stone sampling methods implemented in BEAST. The posterior distribution of trees was summarised into a maximum clade credibility tree. For the largest, CC88-derived dataset a phylogeographic analysis was performed using the Bayesian stochastic search variable selection (BSSVS) model to reconstruct the geographical locations of the ancestral states [42] contained in a sub-sample of the posterior tree distribution of the BEAST runs described above. SpreaD3 v0.9.6 [43] was used to identify statistically well-supported migration routes using the Bayes Factor (BF).

## Genome-wide association analysis (GWAS)

We performed a GWAS analysis in order to find genetic determinants associated to rodent isolates compared to a collection of *S. aureus* whole genome sequences from other hosts. For the latter, we downloaded all RefSeq *S. aureus* genomes with known host (mostly human, bovine, swine and avian) as well as a collection of *S. aureus* genomes representative of the worldwide genetic diversity [5]. The sequence dataset for the GWAS included all rodent sequences (n = 357) with the 50 genetically closest (in terms of *mash* distance [44]) non-rodent sequences to each of them. Identical sequences were removed, and, in order to avoid over-sampling of the CC88 murine cluster, this lineage was further down-sized. This produced a final dataset of 832 sequences (241 rodent and 591 non-rodent controls).

We performed the GWAS analysis using pyseer v1.1.1 [45] from 3 inputs: k-mers, core SNPs and gene presence/absence. For the k-mer-based GWAS analysis, 31 base pair (bp)-long k-mers were generated from the 832 assemblies using *dsk* [46], and summarised using the pyseer script *combineKmers*. A variant calling file (VCF) containing core SNPs was inferred using snippy and snippy-core v4.1 (https://github.com/tseemann/snippy) with the strain RF122 (GCA_000009005.1) as reference. The gene presence/absence matrix was generated using *roary* v3.12.0 [47]. Different runs of pyseer applied two methods for population structure correction (*mash* and phylogenetic distances) and three association models: fixed effects (*SEER*), mixed effects (*FaST-LMM*) and lineage effects (*bugwas*) (**S1 Data**). In all cases we applied a filter *p*-value of 1E-8, a minimum allele frequency of 0.01 and a maximum allele frequency of 0.95.

An initial threshold for the p-value adjusted for population structure in order to consider statistical significance of the k-mers (p < 1.43E-07) was calculated using the script *count_patterns.py* included in the *pyseer* pipeline. Significant k-mers were mapped against a collection of full genome *S. aureus* sequences from a wide variety of clonal complexes in order to identify the gene each k-mer originated from. Genes containing significant k-mers which belonged to the same cluster of orthologous genes (COG), according to *roary*, were grouped. After examination of the gene areas containing the k-mers, only those which involved non-synonymous substitutions were further considered. Finally, only k-mers found in more than one major rodent lineage were analysed, a step taken in order to avoid overrepresentation of the k-mers enriched in the CC88 lineage only. The function of the proteins encoded by these selected genes was predicted using *eggnog* [48].

## Prophage identification

Prophage sequences in genome assemblies were identified using *PHASTER* [49] and pro-phage-associated regions were extracted and annotated using *prokka* v1.13 [34]. In order to establish the genetic relationships between the putative prophages, we calculated *mash* distances [44] between prophage sequences and used them to build a cladogram applying a hier-archical clustering method (function *hclust*) using R [50]. This cladogram was used to explore the presence and distribution of shared prophage sequences across the study population. The identity of these prophages was confirmed by genome-to-genome comparisons using Artemis Comparison Tool (ACT, [51]).

## Prothrombin sequence comparison

Prothrombin protein references were obtained from GenBank for common shrew (XP_004619129.1), rat (NP_075213.2), human (NP_000497.1) and laboratory mouse (NP_034298.1). As no references were readily available for bank vole, common vole and yel-low-necked field mouse, we used publicly available raw sequence reads stored in the Sequence Read Archive (SRA) in order to determine the prothrombin mRNA sequences of these species. In detail, we downloaded the SRA datasets SRR1010821 (bank vole), SRR1325018 (common vole), and SRR9990624 (yellow-necked field mouse) using parallel-fastq-dump and trimmed the raw reads using Trimgalore v0.6.10 (https://github.com/FelixKrueger/TrimGalore) run-ning in automatic adapter detection mode. The trimmed reads were assembled using rnaS-PAdes v3.15.5 [52] and contigs matching prothrombin were selected using diamond blastx v2.1.6 [53].

## Cloning, recombinant protein purification, and coagulation assays

The capacity of several coagulase-like proteins encoded by rodent *S. aureus* isolates to coagu-late plasma from different hosts was tested *in vitro*. Specifically, nine different coagulase-like proteins were selected for cloning of the coding sequences and protein production: core genome-encoded coagulase (Coa) and van Willebrandt factor-binding protein (vWbp) from three strains each (laboratory mouse ST88, bank vole ST49, and human ST8), phage-encoded coagulase-like (Coa') from two strains each (common vole ST3252 and yellow-necked field mouse ST890), and one SaPI-encoded vWbp (SaPI-vWbp) present only in ST3033 strains from wild common shrews. The six *S. aureus* strains were grown overnight in TSB and geno-mic DNA was isolated with the DNeasy Blood & Tissue Kit (Qiagen, Venlo, The Netherlands) according to manufacturer's instructions, except for the addition of lysostaphin during lysis of the cells. The corresponding coagulase-like genes, omitting the information for the signal pep-tide, were amplified via PCR and purified with the NucleoSpin Gel and PCR Clean-up Kit (Macherey-Nagel, Düren, Germany). DNA was inserted into the plasmid pET21b using the NEBuilder HiFi DNA Assembly Master Mix (New England Biolabs, Ipswich, MA, USA) according to manufacturer's instructions, and transformed into chemically competent *Escheri-chia coli* DH5α. The amplified plasmid was isolated using the High Pure Plasmid Isolation Kit (Sigma-Aldrich, Taufenkirchen, Germany) and transformed into chemically competent *E. coli* BL21 (DE3) cells. For protein expression LB broth containing 100 μg/mL ampicillin was inoc-ulated from a preculture to an optical density (OD) of 0.05 and incubated at 37˚C with agita-tion. Afterwards, protein expression was induced by adding Isopropyl-β-D-thiogalactopyranosid (IPTG) to a final concentration of 1 mmol/L. After 3h of additional incu-bation at 37˚C with agitation cells were harvested and lysed with a sonicator. The His-tagged proteins were purified using a HisTrap HP affinity column on an ÄKTA start protein purifica-tion system according to manufacturer's instructions (GE Healthcare, Chicago, IL, USA). The

imidazole-containing elution buffer was changed to PBS using Amicon Ultra Centrifugal Filters (Merck Millipore, Burlington, MA, US) and the protein concentration was determined with a Bradford assay.

To determine the coagulation properties of these proteins, heparinized plasma was isolated from humans, lab rats, lab mice (C57BL/6), and bank voles. Heparinized lab rat plasma (Sprague-Dawley, Genetex, Irvine, CA, USA) and additional heparinized lab mouse plasma (non-Swiss albino, Equitech-Bio, Kerrville, TX, USA) were obtained from a commercial provider. 376 nmol of protein in a volume of 25 µL PBS was added to 300 µL plasma in a 10 mL glass tube and incubated at 37°C without agitation. The coagulation was examined visually at 0.5, 1, 2, 4 and 20 h using a modified coagulation score [54]: 0 = no coagulation; 1 = small clot/flakes; 2 = medium-sized clot; 3 = large clot; 4 = complete coagulation (coagulum sticks to the inverted tube). The scoring was performed in a blinded fashion.

## Results

### Rodent *S. aureus* isolates represent an array of lineages spanning the species diversity

In order to examine the genetic diversity of the rodent *S. aureus* population and their evolutionary origins, we employed a dataset of 1,249 *S. aureus* sequences from the following host species: humans (n = 571), laboratory mice (n = 245), cattle (n = 108), wild voles and mice (n = 43), birds (n = 44), goats (n = 25), lagomorphs (rabbits, hares) (n = 25), pigs (n = 20), laboratory rats (n = 15), sheep (n = 12), wild rats (n = 12), wild red squirrels (n = 13), shrews (n = 5), and others (n = 111)(S1 Table). In total, 245 isolates from laboratory mice belonged to 16 different CCs, predominantly CC88 (127; 51.8%), CC15 (37; 15.1%), CC1 (30; 12.2%) and CC5 (11; 4.5%). The 15 isolates from laboratory rats belonged to CC7 (5; 33.3%), CC20 (3; 20.0%), CC88 (2; 13.3%), CC1 (2; 13.3%), a novel ST (2; 13.3%) and CC8 (1; 6.7%). In contrast, isolates from wild voles and mice (n = 43), wild rats (n = 12) and wild red squirrels (n = 13), belonged to CC49 (29; 43.9%), CC1956 (15; 22.7%), CC890 (9; 13.6%), CC398 (7; 10.6%), CC8 (3; 4.5%), CC130 (2; 3.0%), and CC80 (1; 1.5%), whereas five isolates from wild common shrews sampled in the same geographic region belonged to a single CC (CC3033) consistent with host-specificity (S2 Table). *S. aureus* from 28 urban rats and 7 zoo rodents (guinea pigs, maras and capybaras) belonged to STs typically associated with humans or livestock, including CC398 (12; 34.3%), CC8 (10; 28.6%) and CC97 (4; 11.4%), among others. Maximum-likelihood phylogenetic analysis of all 1,249 *S. aureus* sequences revealed rodent isolates to be interspersed among human and other animal lineages across the tree, consistent with numerous host-switching or spill-over events into rodents during the recent evolutionary history of *S. aureus* (Fig 1). Of note, some rodent-associated lineages represent monophyletic, host-specific clades (Fig 1), which are examined in more detail in the following sections.

### CC88 emerged in laboratory mice after a single host switch followed by wide international dissemination

*S. aureus* CC88 contained the largest rodent-specific phylogenetic cluster observed in our Maximum-Likelihood phylogenetic tree (n = 127), including 124 ST88 isolates and 3 single-locus variants of ST88 (Fig 1). Of the isolates in this clade, 125 were obtained from laboratory mice and 2 from laboratory rats, in 13 different facilities across 6 countries in 4 continents: Canada, France, Germany, Japan, New Zealand and the USA. A CC88-specific Bayesian phylogenetic analysis (Fig 2) included all publicly available sequences belonging to this CC with sampling date and location (n = 209), and allowed us to estimate that the most recent common

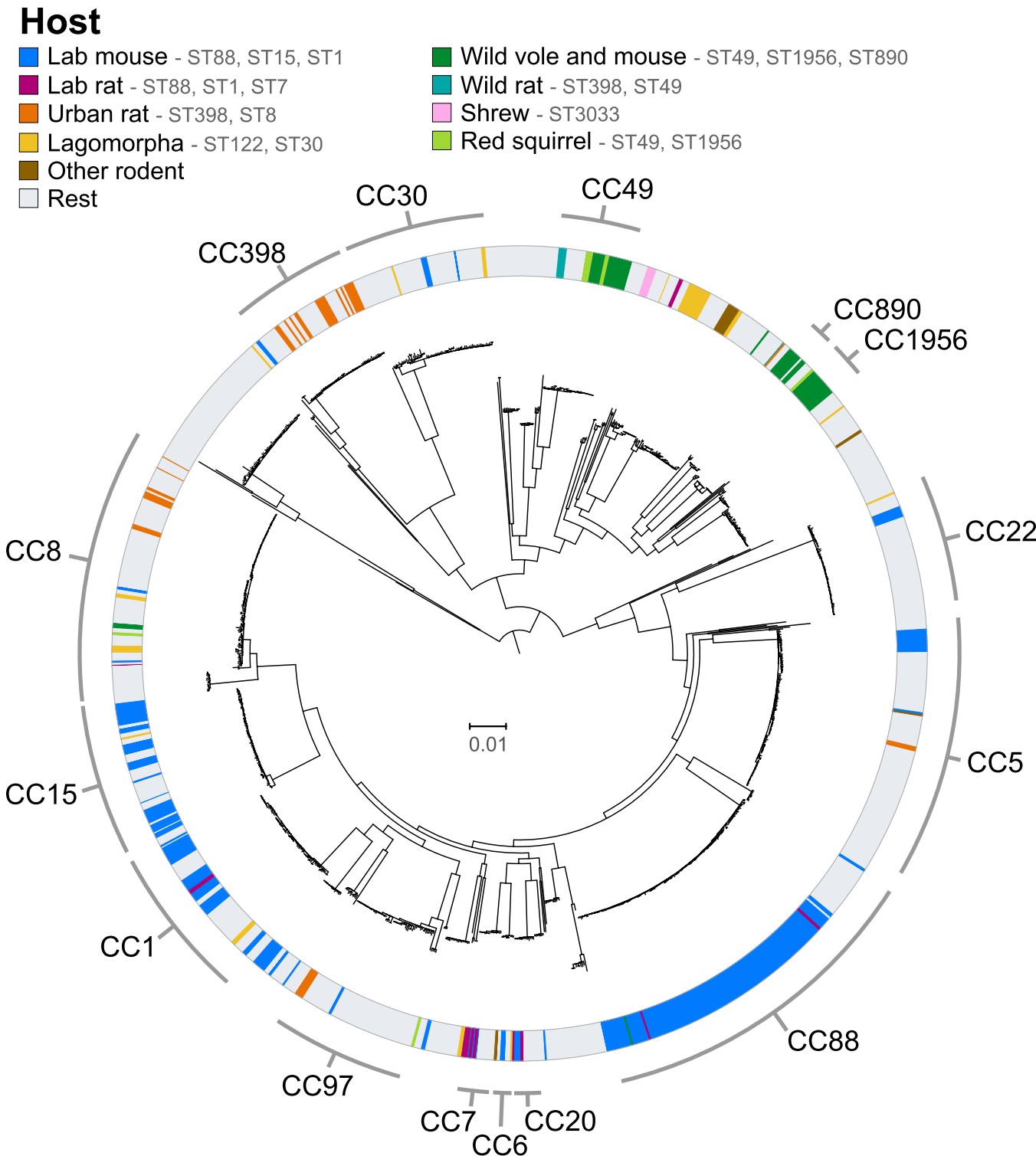

**Fig 1. Rodent *S. aureus* isolates are distributed across the full species diversity.** Maximum likelihood tree of 398 *S. aureus* genomes from rodents and other small mammals together with 851 background sequences from other hosts, mostly humans, representative of the worldwide *S. aureus* genetic diversity. The main clonal complexes among the study population are highlighted.

ancestor (MRCA) of CC88 existed in 1972 (Bayesian Confidence Interval (BCI): 1966.0–1977.5). In addition, phylogeographic reconstruction inferred its most likely origin to have been in South East Asia (Thailand/Cambodia) with a root posterior probability (PP) of 0.32. The CC88 evolutionary rate was determined to be 2.3 (1.92–2.64) × 10⁻⁶ substitutions/site/year. The phylogeny indicates that the murine-associated CC88 clade likely resulted from a single introduction of *S. aureus* into laboratory mice which happened around 1984 (1977.4–1989.4) on the East Coast of North America (**Fig 2**). The subsequent international spread of murine *S. aureus* was driven by the distribution of colonized laboratory mice from a limited number of commercial vendors to research institutes with their own breeding facility around the globe. Although the oldest nodes in the murine cluster exhibited a degree of uncertainty (see **S1 Fig**), the Charles River (CR) facility in Saint-Constant (Canada) was determined to be a possible origin of the lineage (PP = 0.86), after which migration to CR Raleigh (NC, USA) preceded the wider spread to CR facilities and other public and private institutions in North America up to at least 2004 (**Fig 2**). This is supported by the fact that CR Raleigh isolates exhibited the highest genetic diversity and were distributed in groups across the entire murine CC88 lineage. Isolates from CR Kingston (NY, USA) were largely clustered in a single group, suggesting one or few transmissions from CR Raleigh to CR Kingston (between 1990 and 1995). Murine CC88 were also introduced into three research institute-associated breeding facilities in Germany and the phylogeny suggests that these migrations were closely connected,

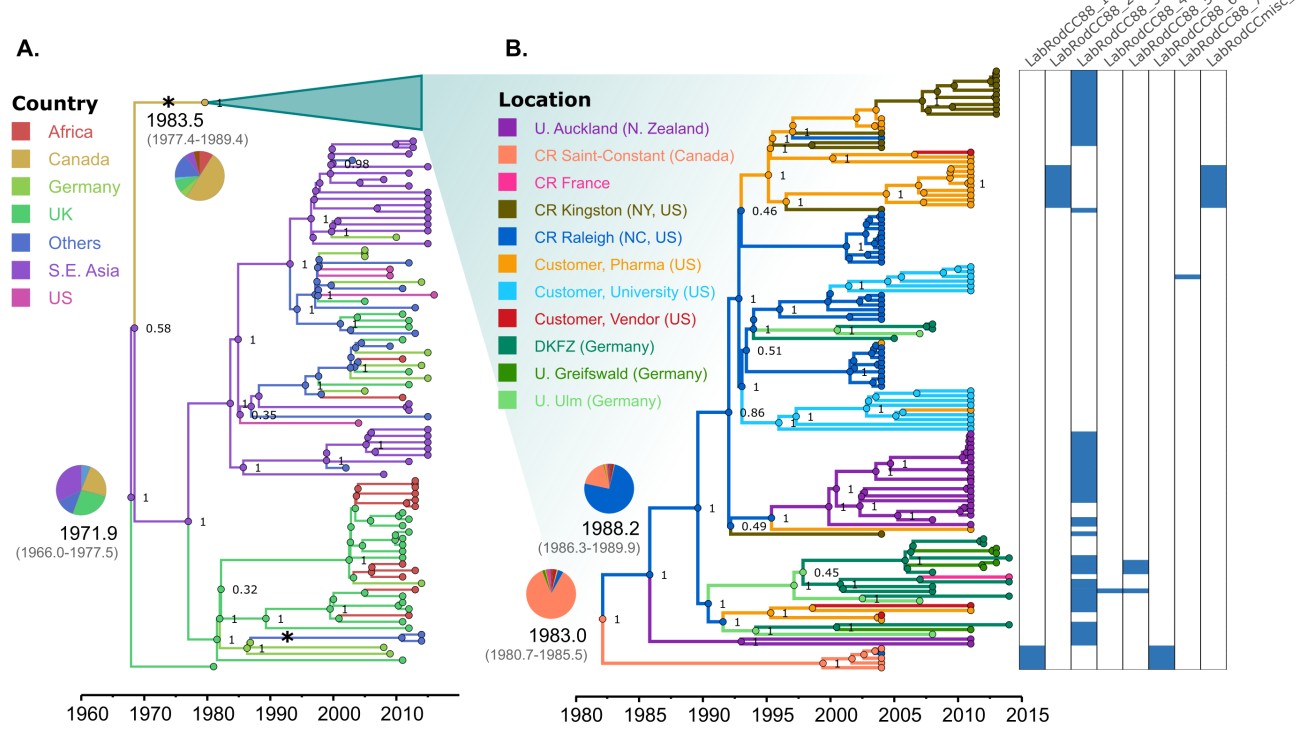

**Fig 2. The murine *S. aureus* CC88 international lineage represents a single introduction into the laboratory mouse population.** Bayesian phylogenetic tree of the *S. aureus* CC88 lineage. Branches and node circles are coloured according to the reconstructed location, and the numbers in key nodes represent statistical support. A) Tree of the full lineage, with the main, international murine cluster collapsed. Asterisks represent host jumps into laboratory mice and rat populations. For the root node and the common node to the murine cluster, the reconstructed dates and the probabilities of each possible location are shown. B) Tree of the international murine cluster, with further locations depending on the sampled institution. The columns at the right-hand side indicate the distribution of the prophages found across laboratory isolates. Abbreviations: CR, Charles River, DKFZ, Deutsches Krebsforschungszentrum, U., University.

possibly because CC88-colonized animals were purchased from the same supplier. Moreover, murine CC88 migrated to a university-associated breeding facility in New Zealand in two events between 1990 and 1995, suggestive of the purchase of breeding pairs from different vendors. Most of these migrations involved *S. aureus* strains that were closely related to strains obtained from USA-based CR customers (vendors, pharmaceutical companies and/or universities), suggesting that CC88-colonized animals originated from the same supplier or were shared between facilities (**Fig 2**).

Of note, we found a second CC88 introduction into the murine population (**Fig 2**), based on isolates from two French laboratory mice that formed a phylogenetic pair (emerged in 2012 (2010.1–2013.1)) clustered within the human CC88 clade that was independent from the international murine CC88 cluster (**Fig 2**).

## Identification of multiple, independent introductions of *S. aureus* into laboratory mice and rats from humans with limited onward dissemination

CC1 and CC15 each contained five rodent-specific phylogenetic clusters representing multiple host jumps from humans and expansions into rodent laboratory populations. Phylodynamic analyses (**S2 Fig**) of these lineages revealed that the oldest of these clusters originated around the year 1980 (1979.2 (1976.3–1982.1) and 1983 (1977.0–1987.8) for the oldest CC15 and CC1 clusters, respectively), in agreement with the reconstructed date of emergence of the international murine CC88 cluster (**Fig 2**). The oldest CC15 cluster included samples from US, Canada and the UK. Subsequently, other CC15 and CC1 clusters emerged during the 1990s including two that also involved samples from different countries (Germany and France in both cases). In addition, singleton monophyletic clusters were detected for many other *S. aureus* lineages (CC5, CC22, CC7, CC188, CC20 and CC6), consistent with frequent spillovers into the laboratory mouse and rat population with limited further expansion.

## Most *S. aureus* from wild rodents in Europe belong to a single lineage CC49

Phylodynamic analysis of CC49 indicated an origin of approximately 1840 with an early split into a clade comprised of swine isolates in Switzerland and a second clade comprised of rodent or human isolates in several European countries (**Fig 3**). The latter clade originated in 1924 (1838.6–1939.5) and comprised of subclades that are largely segregated according to host-species group and geography (**Fig 3**). For example, CC49 sequences from wild rodents segregated according to sampling location (North East Germany, and South West Germany) interspersed with yellow-necked field mouse and bank vole isolates (**Fig 3**). *S. aureus* isolates from red squirrels in the UK Channel Islands were basal to the wild German rodent isolates in the phylogeny (**Fig 3**). Wild rat isolates from Belgium formed a deep-branching cluster indicating a distinct evolutionary history to *S. aureus* in wild mice and voles (**Fig 3**). The remaining *S. aureus* lineages associated with wild voles and mice (CC890, CC1956) and wild shrews (CC3033) were highly host-specific with limited clonal genetic diversity (**Fig 1**). Overall, these data indicate distinct evolutionary histories for *S. aureus* associated with wild and domesticated rodents, respectively.

## Genome-wide association analysis identifies prophage genes as key markers of rodent-host-specificity

The identification of *S. aureus* lineages that have evolved the capacity to colonise and spread in rodent populations prompted us to explore the genetic basis for rodent host-adaptation. Accordingly, we carried out a kmer-based GWAS analysis using *pyseer* to identify genetic

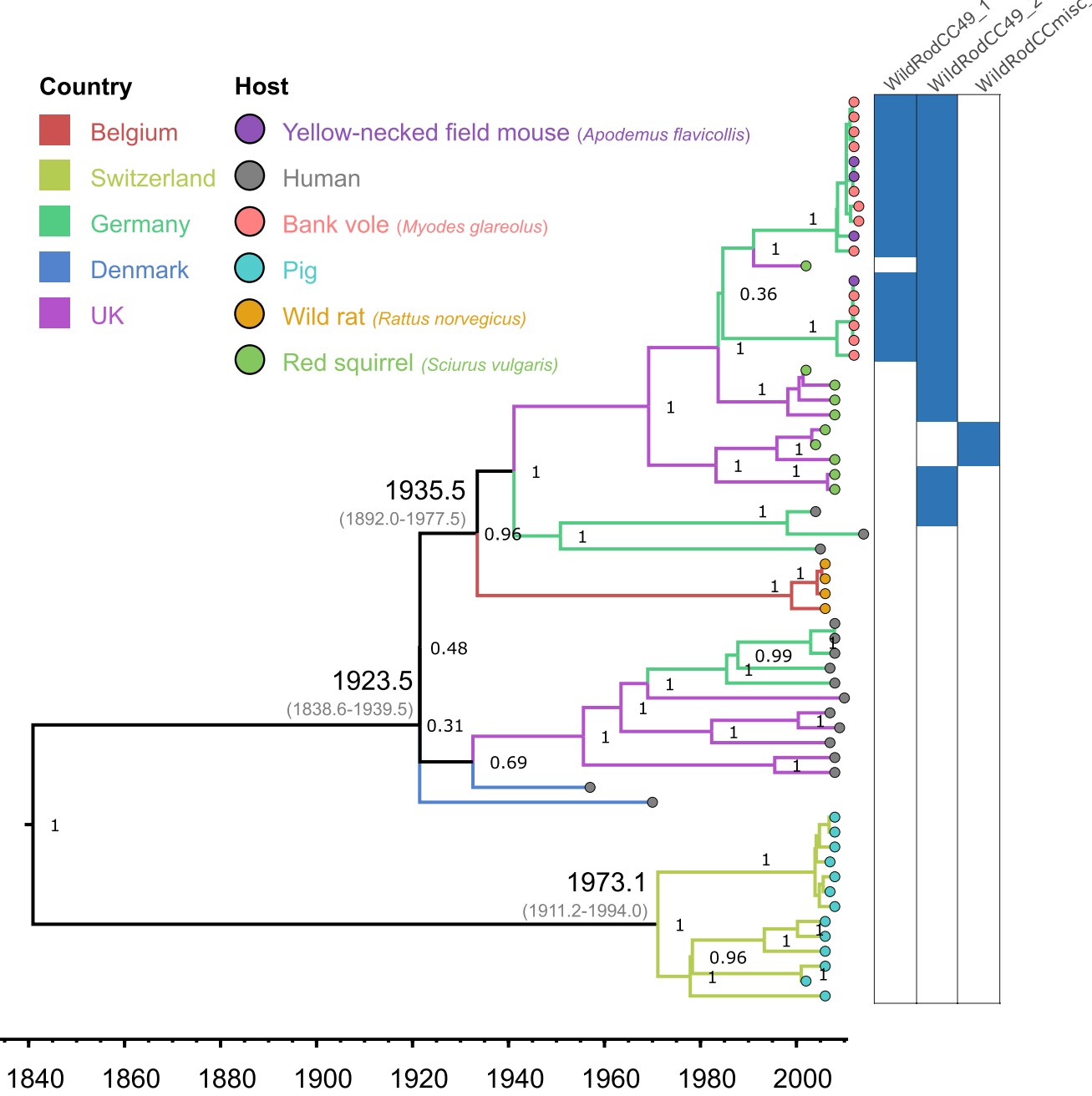

**Fig 3. Time-scaled analysis of *S. aureus* CC49 in wild rodents.** Bayesian phylogenetic tree of the *S. aureus* CC49 lineage. Branch colours show sampling country, and the circles at the tips indicate the bacterial host. The columns at the right-hand side indicate the distribution of three prophages found across wild rodent isolates. The dates for most recent common ancestors of key nodes are shown.

determinants (including whole genes or gene mutations) of *S. aureus* associated with rodent hosts (see Methods section). A total of 41 unique Clusters of Orthologous Groups of proteins (COGs) contained k-mers significantly enriched in rodent *S. aureus* sequences. Of these, the majority (n = 26) were located on phage sequences, including 13 representing enriched whole genes, and 13 representing non-synonymous mutations of phage genes. Of note, phage genes accounted for the 18 most significant hits, indicating that specific prophage sequences were

the best predictors of rodent host-association, and likely play a key role in host adaptation. The 15 GWAS hits found in the chromosome included non-synonymous mutations in genes encoding putative amino acid synthases, transmembrane transporters, 2 superantigen-like proteins (SSL4 and SSL7), IgG-binding protein, a nuclease, a nucleokinase and a helicase (**S3 Table**). Finally, a whole gene encoding a putative restriction enzyme belonging to a type II restriction modification system was enriched in the CC890, CC1956 and CC30 clades. Overall, the relatively small number of GWAS hits out-with MGE indicates that convergent adaptations of different *S. aureus* lineages to the rodent host were rare, implying an important role of lineage-specific traits of host-adaptation, as highlighted previously for bovine *S. aureus* [13].

## Rodent *S. aureus* isolates contain a diverse complement of novel prophages

The identification of phage genes as markers of rodent *S. aureus* led us to examine in detail the diversity and distribution of phages of the family *Siphoviridae* among the rodent *S. aureus* population. A hierarchical clustering tree constructed from pairwise k-mer-based genetic distances of the prophages identified by PHASTER (**S3 Fig**) was used to identify a total of 52 prophage lineages that were present in at least 2 rodent *S. aureus* genomes (see **S4 Table** for a summary of the features of these 52 phage lineages). **Fig 4** depicts the genome maps of the most frequently identified (identified in ≥10 isolates) prophages, and those exhibiting the strongest rodent-specific genetic signatures (see below). *S. aureus* from wild mice and voles contained on average more prophages than isolates from laboratory mice (mean = 2.2 vs 0.6 prophages/genome, respectively; p<0.01). Most (n = 33) prophages belonged to a single CC, associated with a single host-species. However, the remaining 19 prophages were identified in *S. aureus* genomes across lineages and/or host-species. A total of 6 different prophages were shared between *S. aureus* isolates from mice and rats (**S4 Table**). Of note, prophages were not shared between laboratory and wild mice, consistent with the ecological separation of murine *S. aureus* populations in laboratory and wild environments. In contrast, we identified prophages that were shared among isolates from laboratory mice and rats, and among urban rats and companion animals, revealing a distinct pool of MGE in human-associated environments.

Among the prophages, we detected 13 different integrase types (the most common being Sa3int (n = 14)), including 5 novel variants in 10 prophage lineages (8 in wild rodents and 2 in laboratory mice), which where phylogenetically distinct from previously described integrase types [55] (**S4 Fig**), and which exhibited novel chromosomal integration sites. BLAST analysis of the 52 full prophage sequences revealed that 28 phages were novel, i.e., did not have a match in the GenBank sequence database with ≥90% coverage and ≥90% nucleotide sequence identity. All but one (16 of 17, 94.1%) of those identified in wild rodents were novel, whereas only 12 of 35 (34.3%) phages identified in laboratory rodents were novel, highlighting the unique nature of phages associated with wild rodent isolates. Importantly, 32 of the 52 phage lineages contained at least one of the rodent-associated hits identified by GWAS (**Fig 4**). Most hits were associated with modules for DNA replication, DNA packaging and head morphogenesis and tail. However, a gene encoding a novel serine protease, named *jep* (JSNZ extracellular protease), was identified in 4 different phages in both laboratory and wild rodents, and in 12.4% of the rodent genomes (and only 0.5% of the non-murine *S. aureus* genomes). Taken together, we have identified a unique complement of phages associated with rodent *S. aureus* with distinct pools of phages circulating in wild and laboratory populations, respectively.

## Rodent-specific phages encode novel determinants of host-specificity

An array of novel factors predicted to be involved in host-pathogen interactions were encoded among rodent *S. aureus* phages. Namely, genes for a putative serine protease, superantigens,

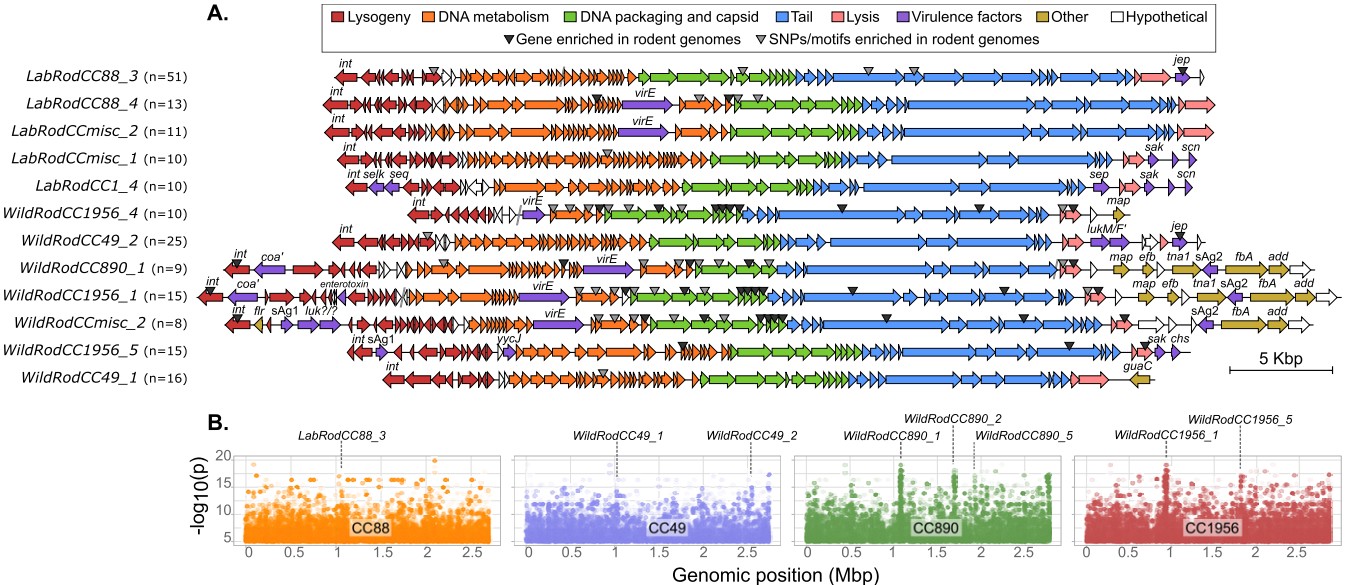

**Fig 4. The GWAS analysis reveal bacteriophages are the main source of rodent-specific *S. aureus* genetic variation.** A) Genome maps of the most frequent prophages and those with the highest number of rodent-specific genetic signatures. Genes are coloured according to their function (see legend at the top). Triangles highlight phage genes harbouring rodent-specific genetic signatures (i.e. GWAS hits). Virulence factors and other genes are labelled. B) Manhattan plots showing the GWAS hits across representative rodent *S. aureus* genomes belonging to the main four lineages found: CC88, CC49, CC890 and CC1956. Peaks with the highest statistical significance correspond to strain-dependent phages as indicated. add: adenosine deaminase; chs: chemotaxis-inhibiting protein (CHIPS); coa': putative coagulase-like protein; efb: extracellular fibrinogen-binding protein; flr: formyl peptide receptor (FPR)-like 1 inhibitory protein; fba: putative Fg-binding protein; guaC: GMP reductase; int: integrase; jep: JSNZ extracellular protease; luk?/?: undetermined leukocidin; lukM/F': leukocidin MF'; map: major histocompatibility complex class II analog protein; sAg1: putative superantigen 1; sAg2: putative superantigen 2; sak: staphylokinase; scn: staphylococcal complement inhibitor (SCIN); selk: staphylococcal enterotoxin-like protein K; sep: staphylococcal enterotoxin P; seq: staphylococcal enterotoxin Q; tna1: tryptophanase; virE: virulence-associated protein E; yycJ: Metallo-beta-lactamase family protein.

superantigen-like proteins and coagulase-like proteins were distributed among phages in distinct *S. aureus* CCs (**Figs 2B, 3 and 4**). In particular, the *jep* gene encoding a novel protease is located on a prophage (*LabRodCC88_3)* distributed across multiple subclades of the globally distributed CC88 laboratory mouse cluster (**Fig 2B**). Furthermore, the *WildRodCC49_2* and *WildRodCCmisc_1* phages, which contain their own allelic variant of the *jep* gene, were identified in all isolates within the wild rodent CC49 lineage (**Fig 3**) suggesting a role in the adaptation of CC49 to wild rodents.

Another determinant of host-specificity is the pore-forming toxin LukMF', a member of the family of bicomponent leucocidins that includes the Panton-Valentine leukocidin (PVL) [56]. lukMF' has recently been implicated in contributing to fulminant skin and soft tissue infections in wild squirrels [30]. LukMF' was encoded by the *phage WildRodCC49_2*, common to wild mice (n = 17) and squirrel (n = 5) isolates (**S2** Table).

Moreover, phages containing genes encoding allelic variants (72% aa identity) of a putative coagulase-like protein (*coa*') were present in all CC890 (*WildRodCC890_1*, in wild mice) and CC1956 (*WildRodCC1956_1*, in wild mice and red squirrels) isolates. Unusually for accessory genes, the coagulase-like protein genes were located adjacent to the phage integrase gene (**Fig 4**). Previous studies have demonstrated that *S. aureus* produces several secreted proteins with plasma coagulase activity [57]. In particular, *S. aureus* produces coagulase (Coa) and von Willebrand factor-binding protein (vWbp) which can activate host prothrombin and form fibrin clots, thereby promoting abscess formation [57,58]. The identification of a large array of novel putative coagulase proteins encoded by rodent *S. aureus* is striking and suggests an important role in host adaptation (**S5 Table**). Accordingly, we aimed to test the ability of the encoded

proteins to coagulate plasma from humans and rodents *in vitro*. We generated recombinant proteins of the core genome-encoded coagulase (Coa) and vWbp from human-derived ST8, laboratory mouse-derived ST88, and wild rodent-derived ST49 strains, as well as phage-encoded coagulase-like proteins (Coa') from wild rodent-derived ST3252 and ST890 strains, and one SaPI-encoded vWbp-like protein (vWbp') identified only in ST3033 strains from wild shrews. While Coa encoded by ST8 and ST88 were closely related (90.4% aa identity), ST49 Coa and the phage-encoded Coa' variants differed markedly from the Coa ST8 (68.8, 29.3 and 40.1% identity for Coa ST49, Coa' ST3252 and Coa' ST890, respectively) (S5 Fig). Similarly, vWbp sequences from rodent and shrew-derived *S. aureus* isolates differed markedly from the vWbp sequence of the human-adapted ST8 lineage (S5 Fig). For both, Coa and vWbp, the greatest sequence variation was observed in the domains required for prothrombin-binding (D1 and D2). All nine recombinant proteins were tested for coagulation activity on plasma obtained from human, laboratory mouse, laboratory rat and bank voles (**Fig 5**).

Overall, all proteins tested coagulated plasma from at least one host-species. Notably, coagulation activity was highly species-dependent, suggesting a role for coagulases in host adaptation (**Fig 5**). The core genome-encoded Coa from human ST8, murine ST88 and ST49 strongly coagulated human plasma, but exhibited no or neglectable activity on mouse, rat and bank vole plasma. In contrast, the phage-encoded Coa' variant from yellow-necked field mouse-derived ST890 strongly coagulated bank vole, mouse and rat plasma, but exhibited low level activity for human plasma. Similarly, the core genome-encoded vWbp from human ST8 potently coagulated human plasma as well as blank vole plasma, but exhibited negligible activity for lab mouse and lab rat plasma. Core genome-encoded vWbp from ST88 and ST49 had low activity for human plasma and negligible activity for mouse and rat plasma. Strikingly, the SaPI-encoded vWbp variant (vWbp' ST3033) from shrews potently coagulated bank vole plasma, exhibited moderate activity for mouse and rat plasma and none for human plasma.

The host-dependent functionality of different coagulases likely reflects distinct tropisms for the substrate prothrombin which differs in sequence in different host-species [59–62]. The human prothrombin heavy chain interacts with Coa via two binding sites, the Trp148 binding pocket and exosite 1 [63]. Both Coa and vWbp allosterically activate prothrombin through insertion of the first two N-terminal residues of Coa/vWbp into the activation pocket on prothrombin [63,64]. The prothrombin heavy chain has 81% aa identity with the prothrombin of mice (*Mus musculus*), 79% with rats (both *Rattus rattus* and *Rattus norvegicus*), and less than 75% with common shrews (*Sorex araneus*) (**S6 Fig**). The interacting amino acid residues in the Coa binding sites and activation pocket were conserved between human and mouse/vole prothrombin sequences, but the shrew prothrombin displayed a R75→K mutation in the exosite 1. In addition, mouse, vole and shrew prothrombin displayed sequence variations adjacent to the binding pocket and the activation pocket (**S6 Fig**). These prothrombin modifications likely drive selection for new host-specific coagulase variants.

Overall, these data indicate that *S. aureus* chromosomally-encoded secreted coagulases had limited functional activity for most rodent plasmas whereas coagulases encoded on rodent-specific MGE could mediate coagulation of rodent plasma suggesting a key role for these elements in murine host-adaptation.

## Discussion

Addressing the ongoing threat of emerging infectious diseases affecting humans and animals requires identification of the main potential reservoirs for new pathogens, and an understanding of how they evolve and spread within new host-species populations. Recent bacterial typing studies have reported an array of *S. aureus* STs that are associated with laboratory rodent

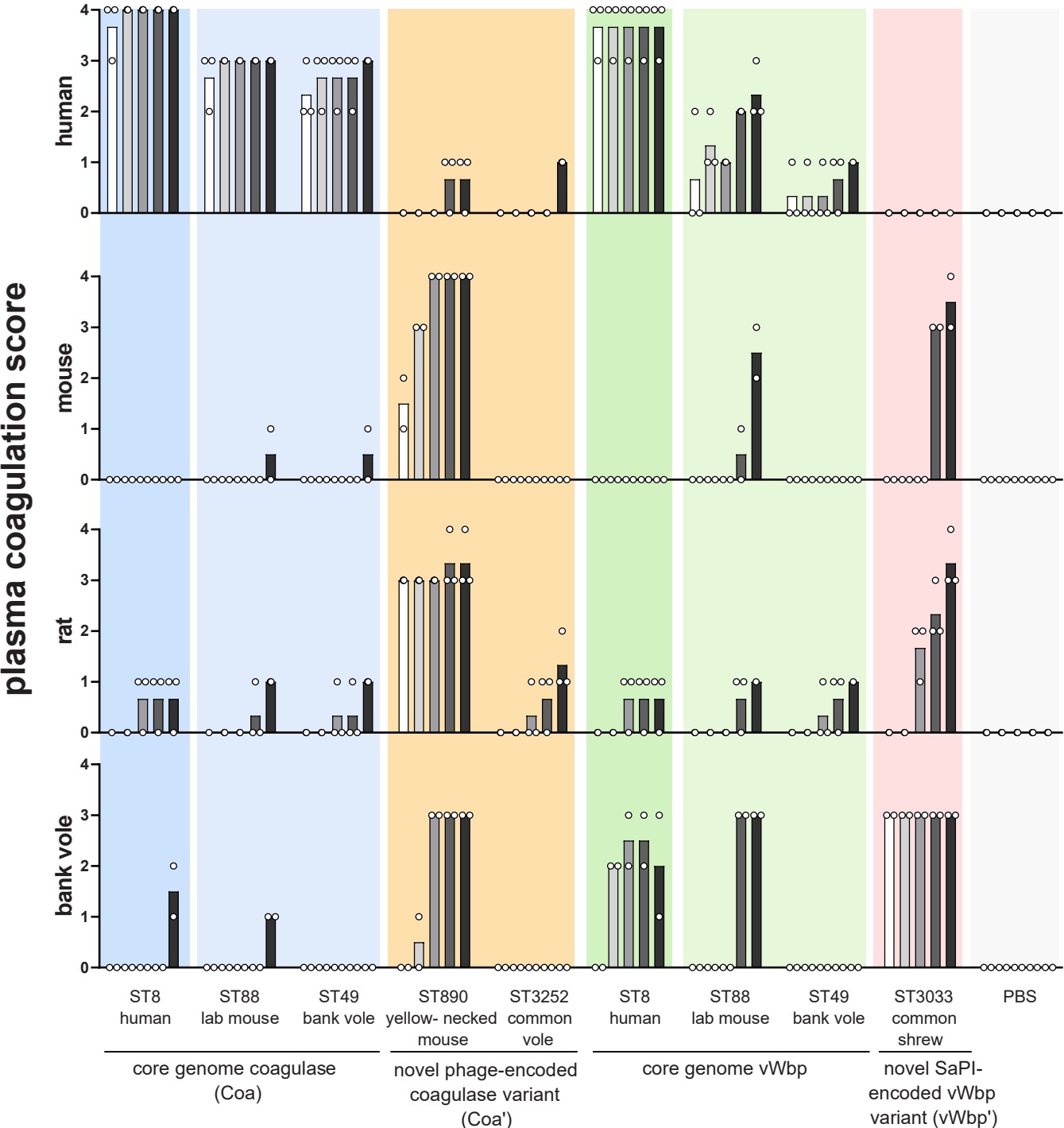

**Fig 5. Rodent- and shrew-derived *S. aureus* coagulase-like proteins exhibit host-specific coagulation.** Coagulation of human, rat, mouse and bank vole plasma was induced by different recombinant *S. aureus* coagulase-like proteins from human, lab mouse, bank vole (*Myodes glareolus*), yellow-necked field mouse (*Apodemus flavicollis*), common vole (*Microtus arvalis*), and common shrew (*Sorex araneus*). Heparinized plasma from different species was mixed with *S. aureus* coagulase-like proteins and coagulation was visually assessed after 0.5, 1, 2, 4 and 20 h (indicated by individual bars with increasing grey scale). For the experiments in human plasma, plasma from 3 different donors was tested. For the experiments in animal plasma, plasma pools were used and independently tested two (mouse and bank vole) or three (rat) times. The bars show the respective mean value.

populations around the world [16,17]. Here, using a phylogenomic approach, we report that these lineages are broadly representative of the *S. aureus* species diversity. Many cases of *S. aureus* isolation from rodents likely involved sporadic human-to-rodent spillovers with limited levels of onward transmission, reflecting the nature of laboratory rodent husbandry. However, we identified several monophyletic, rodent-specific populations which have arisen via single host-switching events presumably from humans, followed by widespread intercontinental dissemination.

In particular, *S. aureus* CC88 represents a large, internationally spread lineage, found only in laboratory rodents. Our phylogenetic data suggest that it most likely emerged in CR facilities in Canada, from which it spread to facilities in the US East Coast and subsequently (from 1990) throughout the world. Several factors should be considered when interpreting the phylo-dynamic data for *S. aureus* from laboratory rodents: hygiene measures and caging effectively prevent horizontal transmission between barrier rooms, but still enable vertical transmission from breeding pairs to offspring [16]. In addition, while vendors are active on a global scale, shipping *S. aureus*-colonized mice between continents, actual animal movements were not documented and thus intermediate steps (e.g. other vendors or users) may be missing in our study. Thus, the locations suggested to be involved in migration events should be interpreted with caution. Nevertheless, it is likely that there is a human origin for CC88 which has been largely reported in humans in South East Asia and sub-Saharan Africa but not typically in other parts of the world [18]. CC88 emerged in laboratory mice around 1983, within a few years of the oldest CC1 and CC15 murine clusters, and these contemporaneous events may reflect animal handling or biosafety practices that were in place during the late 1970s-80s. At that time, barrier to barrier transfer of laboratory mice was still common practise. Of note, only limited numbers of transmission events into mice have been detected subsequently, likely reflecting stricter regulations regarding biosafety. Finally, the narrow geographic sampling of wild mouse, vole, shrew, and guinea pig isolates limits the scope of the phylogeographic analysis.

Some human *S. aureus* CCs have a particular propensity to undergo host-switches, adapt, and spread in mouse populations. For example, we observed several spillover events for the *S. aureus* lineages CC1 and CC15, which are commonly found in humans.

As previously reported [26], wild rodents (mice, voles, red squirrels and rats) were colonized by *S. aureus* lineages distinct to those found in laboratory mice, dominated by CC49, CC890 and CC1956. In particular, CC49 has been reported to be a cause of fatal exudative dermatitis in UK squirrels and it is speculated that the expression of the lukMF' toxin may be associated with the lethal infections of squirrels following spillover events from a reservoir species (such as bank voles or rats) with overlapping habitats [29,30]. LukMF' is commonly found in ruminant and rodent *S. aureus* isolates [65–67]. It lyses bovine, ovine, goat and mouse neutrophils *in vitro* by binding to chemokine receptor CCR1, but has no cytolytic effect on human neutrophils, which lack CCR1 [68].

Our analysis did not detect evidence of convergent host-adaptation between lineages, and genetic diversification was largely lineage-dependent, consistent with previous findings for *S. aureus* associated with host-species [5,13]. However, we identified a large and diverse family of novel *S. aureus* prophages that are associated with a rodent host ecology. The majority of prophages identified among wild rodent *S. aureus* were not represented in public sequence repositories, encoded novel integrase variants and proteins putatively involved in host-pathogen interactions. For instance, CC1 and CC15 isolates from laboratory mice were marked by the loss of human-associated prophages and acquisition of murine-associated prophages consistent with adaptation to the murine host. Moreover, the remarkable success of the CC88 lineage has not been observed for other lineages and we hypothesise that the expansion was aided by

the acquisition of prophages such as one (*LabRodCC88_3*) that encoded the novel serine protease JEP. This protease was also encoded by unrelated prophages found in CC49 isolates from wild rodents (*WildRodCC49_2* and *WildRodCCmisc_1*, in wild mice, voles, and red squirrels), indicative of a key role in adaptation of *S. aureus* to both laboratory and wild rodents. Overall, this suggests that the emergence and expansion of *S. aureus* in rodent populations is driven by bacteriophage-encoded effectors.

To investigate this further we analyzed phage- and SaPI-encoded coagulase-like proteins and demonstrated that they conferred the ability to mediate coagulation of rodent plasma demonstrating a role for acquisition of MGE in adaptation to rodent hosts. Previously, we reported that novel variants of SaPI-encoded vWbp identified among sheep *S. aureus* and *S. aureus* subsp. *anaerobius*, the causative agent of Morels' disease, had ruminant-specific functional activity [9,69]. Furthermore, different coagulase-positive staphylococcal species have evolved coagulases with a wide spectrum of host-tropisms [61]. These studies highlight the key importance of plasma coagulation for *S. aureus* survival *in vivo* and the strong selective pressure driving the diversification of coagulases to promote abscess formation and persistence in different host-species.

In conclusion, we used a combination of population comparative genomics, phylodynamics and GWAS to reveal the genomic diversity of *S. aureus* isolates found in rodent populations. *S. aureus* isolates from laboratory mice resulted from numerous human-to-mouse host jump events, including a CC88 clade that disseminated worldwide via introductions from commercial mouse providers. In contrast, wild rodents contained a distinct set of *S. aureus* lineages consistent with their independent ecological and evolutionary history. A unique set of diverse prophages were identified to be circulating among *S. aureus* in rodent populations, some of which encoded novel determinants of host-specificity that likely promote colonization and persistence. Our findings highlight the remarkable promiscuity of *S. aureus* and its capacity to adapt to different niches and expand into new host-species populations. Our findings reinforce the importance of employing a One Health approach to investigate the dynamics of bacterial pathogens at the interface between humans, and domesticated and wild animals.

## Supporting information

**S1 Fig. Densitree of Bayesian posterior distribution of CC88 international murine cluster.** The plot shows all possible topologies, parts of the tree with a higher topology agreement look sharper, whereas areas with more uncertain topologies look more blurred.
(DOCX)

**S2 Fig. CC1 and CC15 murine-specific clusters extracted from the phylogenetic Bayesian trees of both *S. aureus* lineages.**
(DOCX)

**S3 Fig. Hierarchical clustering tree built from pairwise genetic distances (calculated with Mash) between prophage sequences (as predicted by PHASTER) across the whole study dataset.** Tips names and circles are coloured according to the integrase type detected through BLAST.
(DOCX)

**S4 Fig. Maximum-likelihood phylogenetic analysis of the integrases in rodent prophages.** Reference sequences for each integrase type are coloured in cyan. Putative novel integrase types, genetically different and showing different integration sites, are coloured in orange.
(DOCX)

**S5 Fig. Protein sequence variation among core genome- and MGE-encoded coagulases.** A) Amino acid-based identity matrix for core-genome encoded coa (human-derived ST8, laboratory mouse-derived ST88, and bank vole-derived ST49 strains) as well as phage-encoded vWbp variants from yellow-necked field mice (Coa' ST980) and common voles (Coa' ST3252). B) Alignment of core genome- and phage-encoded coagulases. The prothrombin-binding domains D1 and D2 and the fibrinogen-binding domain are indicated. Amino acid residues interacting with the exosite 1 on prothrombin are highlighted. The highlighted N-terminal amino acids induce allosteric activation of prothrombin by inserting into the prothrombin activation pocket. C) Amino acid-based identity matrix for core-genome encoded vWbp (human-derived ST8, laboratory mouse-derived ST88, and bank vole-derived ST49 strains), as well as SaPI-endoded vWbp' (common shrew-derived ST3033). D) Alignment of core genome- and SaPI-encoded vWbps.The N-terminal peptide, the prothrombin-binding domains D1 and D2 and the van Willebrand factor-binding domain are indicated. (DOCX)

**S6 Fig. Protein sequence variation among human, rodent and shrew prothrombin.** A) Amino acid-based identity matrix for the prothrombin heavy chain derived from humans, laboratory mice, yellow-necked field mice, rats, bank voles and common voles. B) Alignment of human, rodent and shrew prothrombin sequences (heavy chain). The two Coagulase binding sites, Trp148 binding pocket and exosite 1, are indicated. Both Coa and vWbp allosterically activate prothrombin through insertion of their first two N-terminal residues into activation pocket on prothrombin. (DOCX)

**S1 Data. Outputs of different GWAS analyses.**
(XLSX)

**S1 Table. Sequence datasets employed in the study.**
(XLSX)

**S2 Table. Genome sequences and associated metadata.**
(XLSX)

**S3 Table. GWAS hits not associated with phages.**
(XLSX)

**S4 Table. Summary of features of 52 identified phages.**
(XLSX)

**S5 Table. List of identified genes predicted to encode coagulases.**
(XLSX)

## Author Contributions

**Conceptualization:** Silva Holtfreter, J. Ross Fitzgerald.

**Formal analysis:** Gonzalo Yebra, Daniel Mrochen, Stefan Fischer, Florian Pfaff, Rainer G. Ulrich, Kathleen Pritchett-Corning, Silva Holtfreter, J. Ross Fitzgerald.

**Funding acquisition:** Silva Holtfreter, J. Ross Fitzgerald.

**Supervision:** Stefan Fischer, Rainer G. Ulrich.

**Writing – original draft:** Gonzalo Yebra, Silva Holtfreter, J. Ross Fitzgerald.

**Writing – review & editing:** Daniel Mrochen, Silva Holtfreter, J. Ross Fitzgerald.

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
