## [Decision Letter · Decision Letter 0]

13 May 2024

Dear Prof. Fitzgerald,

Thank you very much for submitting your manuscript "Bacteriophage-driven emergence and expansion of Staphylococcus aureus in rodent populations" for consideration at PLOS Pathogens. As with all papers reviewed by the journal, your manuscript was reviewed by members of the editorial board and by several independent reviewers. The reviewers appreciated the attention to an important topic. Based on the reviews, we are likely to accept this manuscript for publication, providing that you modify the manuscript according to the review recommendations.

Please address the issues related to sampling that were raised by both reviewers, and the GWAS issues indicated by reviewer 2. In addition, the posterior probabilites are not high in some of your phylogeographic and phylogenetic analyses, which needs to be better acknowledged. For example, analysis in Figure 2A may best support an origin of CC88 in SE Asia, but it does so at only 32% PP - hardly convincing. When looking at the geographic origin of murine CC88 in Figure 2A the pie-chart seems to only support Canada at about 50% PP, but in the text describing Figure 2B you indicate the support is 86% (probably should also show pie-charts in 2B). The Figure 2B result is better, but why the difference? Is it sampling related? Based on the analysis presented, you should be more circumspect in the Abstract about where this murine lineage may have emerged. In Figure 3, the CC49 node that includes human and rodent isolates has weak support, possibly due to a closely timed radiation of lineages?

Sincerely,

D. Ashley Robinson, Ph.D.

Academic Editor

PLOS Pathogens

Helena Boshoff

Section Editor

PLOS Pathogens

Michael Malim

Editor-in-Chief

PLOS Pathogens

orcid.org/0000-0002-7699-2064

Please address the issues related to sampling that were raised by both reviewers, and the GWAS issues indicated by reviewer 2. In addition, the posterior probabilites are not high in some of your phylogeographic and phylogenetic analyses, which needs to be better acknowledged. For example, analysis in Figure 2A may best support an origin of CC88 in SE Asia, but it does so at only 32% PP - hardly convincing. When looking at the geographic origin of murine CC88 in Figure 2A the pie-chart seems to only support Canada at about 50% PP, but in the text describing Figure 2B you indicate the support is 86% (probably should also show pie-charts in 2B). The Figure 2B result is better, but why the difference? Is it sampling related? Based on the analysis presented, you should be more circumspect in the Abstract about where this murine lineage may have emerged. In Figure 3, the CC49 node that includes human and rodent isolates has weak support, possibly due to a closely timed radiation of lineages?

Reviewer Comments (if any, and for reference):

Reviewer's Responses to Questions

**Part I - Summary**

Reviewer #1: Thank you for the opportunity to review “Bacteriophage-driven emergence and expansion of Staphylococcus aureus in rodent populations” by Yebra and colleagues. The authors present a fascinating study that includes population genomics, phylodynamic, and genome-wide association analysis of S. aureus from rodent populations to trace the evolutionary history and investigate for host-adaptation. The results coalesce into an intriguing story of rodent-associated lineages and host-adaptation through bacteriophage acquisition. In particular, the identify ST88 as a rodent-associated lineage that is likely associated with laboratory rodents. They also identify a number of previously-unreported S. aureus phages, which the subsequently found to encode host-specific proteases. This is one of the few examples of where the evolutionary impact of temperate bacteriophages are revealed.

Overall, their analysis was an excellent combination of genomic and molecular approaches. Their methods are clear, comprehensive, and easy to follow. In addition, the use and incorporation of data from Staphopia presents an excellent use case for the repository. The figures are aesthetically pleasing and informative.

In terms of critiques, I have none that are substantial. Usually in similar papers there are shortcomings in the phylodynamic analysis, but here the authors are thorough with model testing and also present their limitations. The only minor comment is that it may be worth mentioning that the limited geographic sampling of wild mouse, vole, shrew, and guinea pig isolates limits the phylogeographic analysis (i.e., you can’t infer ancestral locations from unsampled locations). However, the discussion mentions something to this effect. In supplemental figure 3, you could probably drop the tip labels and just use shapes since you can’t read the text, but again, that is a minor stylistic critique. I really enjoyed reading the publication and think that it will be of broad interest to the microbial genomics, genomic epidemiology, and molecular biology readership.

Reviewer #2: This population genomics study looked at the evolutionary history of rodent-adapted Staphylococcus aureus using a combination of phylodynamics and genome-wide association studies (GWAS). The authors describe the emergence of CC88 as a laboratory mice clone in the ~1980 as well as several limited introductions of S. aureus into laboratory rodents, while distinct clones are found in wild rodents. Finally, they identify an association between mice adaptation and the presence of a novel phage. This association was further investigated by assessing the host-specificity of the recombinant coagulase proteins obtained from the core genome vs. phage-encoded coagulases.

The manuscript is well written, and the genomics methods are state-of-the art. The conclusions are supported by the genomic analysis and the coagulase activity testing, however, no major new findings emerge: the CC88 has been previously identified and the association between presence of a phage and host adaptation has been reported in previous GWAS with a similar design.

**Part II – Major Issues: Key Experiments Required for Acceptance**

Reviewer #1: None

Reviewer #2: 1. Sampling is key in this kind of studies, yet little is said on how the isolates were collected. How were they obtained? Were they selected somehow?

2. Multiple GWAS approaches were used (fixed effect models, linear mixed model, lineage effects), yet it is unclear how the output of the approaches overlaps (or not) and how the final 41 hits were selected (i.e. just one approach? Intersection of multiples approaches?). I recommend providing all GWAS results in the supplementary data, including non-significant hits.

**Part III – Minor Issues: Editorial and Data Presentation Modifications**

Reviewer #1: Minor changes to supplemental figures, but not required

Reviewer #2: 1. Please add line numbers to the manuscript

2. Please provide the heritability estimated calculated by Pyseer

3. From figure 4B it looks like the GWAS was done on a per-CC basis, however this is not clearly said in the results: please clarify.

4. Table S1: please list all genomes included in the analysis

5. “Overall, the relatively small number of GWAS hits out-with MGE indicates that convergent adaptations of different S. aureus lineages to the rodent host were rare”: this sentence is unclear to me

PLOS authors have the option to publish the peer review history of their article (what does this mean?). If published, this will include your full peer review and any attached files.

Reviewer #1: **Yes: **Taj Azarian

Reviewer #2: No

Figure Files:

Data Requirements:

Reproducibility:

References:

---

## [Editor Report · Decision Letter 1]

27 Jun 2024

Dear Prof. Fitzgerald,

We are pleased to inform you that your manuscript 'Bacteriophage-driven emergence and expansion of Staphylococcus aureus in rodent populations' has been provisionally accepted for publication in PLOS Pathogens.

Best regards,

D. Ashley Robinson, Ph.D.

Academic Editor

PLOS Pathogens

Michael Wessels

Section Editor

PLOS Pathogens

Michael Malim

Editor-in-Chief

PLOS Pathogens

orcid.org/0000-0002-7699-2064
---

## [Editor Report · Acceptance letter]

15 Jul 2024

Dear Prof. Fitzgerald,

We are delighted to inform you that your manuscript, "Bacteriophage-driven emergence and expansion of Staphylococcus aureus in rodent populations," has been formally accepted for publication in PLOS Pathogens.

Best regards,

Michael Malim

Editor-in-Chief

PLOS Pathogens

orcid.org/0000-0002-7699-2064